# Neuroinflammation in Alzheimer’s Disease: A Potential Role of Nose-Picking in Pathogen Entry via the Olfactory System?

**DOI:** 10.3390/biom13111568

**Published:** 2023-10-24

**Authors:** Xian Zhou, Paayal Kumar, Deep J. Bhuyan, Slade O. Jensen, Tara L. Roberts, Gerald W. Münch

**Affiliations:** 1NICM Health Research Institute, Western Sydney University, Westmead, NSW 2145, Australia; p.zhou@westernsydney.edu.au (X.Z.); d.bhuyan@westernsydney.edu.au (D.J.B.); 2Pharmacology Unit, School of Medicine, Western Sydney University, Campbelltown, NSW 2560, Australia; 20645276@student.westernsydney.edu.au; 3Ingham Institute for Applied Medical Research, Liverpool, NSW 2170, Australia; s.jensen@westernsydney.edu.au (S.O.J.); tara.roberts@westernsydney.edu.au (T.L.R.); 4Microbiology and Infectious Diseases Unit, School of Medicine, Western Sydney University, Liverpool, NSW 2170, Australia; 5Oncology Unit, School of Medicine, Western Sydney University, Liverpool, NSW 2170, Australia

**Keywords:** Alzheimer’s disease, neurodegeneration, neuroinflammation, nose-picking, olfaction, hand hygiene, brain, bacteria, viruses, fungi

## Abstract

Alzheimer’s disease (AD) is a complex neurodegenerative disorder characterized by progressive cognitive decline and memory impairment. Many possible factors might contribute to the development of AD, including amyloid peptide and tau deposition, but more recent evidence suggests that neuroinflammation may also play an—at least partial—role in its pathogenesis. In recent years, emerging research has explored the possible involvement of external, invading pathogens in starting or accelerating the neuroinflammatory processes in AD. In this narrative review, we advance the hypothesis that neuroinflammation in AD might be partially caused by viral, bacterial, and fungal pathogens entering the brain through the nose and the olfactory system. The olfactory system represents a plausible route for pathogen entry, given its direct anatomical connection to the brain and its involvement in the early stages of AD. We discuss the potential mechanisms through which pathogens may exploit the olfactory pathway to initiate neuroinflammation, one of them being accidental exposure of the olfactory mucosa to hands contaminated with soil and feces when picking one’s nose.

## 1. The Role of Neuroinflammation in Alzheimer’s Disease

Alzheimer’s disease (AD) is the most common form of dementia, affecting more than 55 million people worldwide in 2022 [1]. The most histologically apparent hallmarks of AD include the accumulation of amyloid-beta (Aβ) peptide forming senile plaques and hyperphosphorylated tau protein forming neurofibrillary tangles in the brain. These protein deposits are thought to be part of pathological cascades, leading to neuronal dysfunction and cognitive decline [2]. Aβ, which is a crucial marker of senile plaques in AD patients and links infection to plaque formation, an important characteristic of AD, can be produced through the cleavage of the APP protein [3].

Neuroinflammation, characterized by the activation of microglia and astrocytes and the release of pro-inflammatory cytokines and free radicals, has been increasingly recognized as a significant contributor to AD pathogenesis and progression [4]. One of the activators of microglia is Aβ, which binds to the Receptor for Advanced Glycation End products, a cell surface receptor that is involved in inflammation and immune responses. The activation of RAGE by Aβ may contribute to microglial activation [5,6].

There is even some evidence to suggest that Aβ may have antibacterial properties as a defense mechanism against microbial infections in the brain. For example, Aβ can form aggregates that entrap and neutralize bacteria in mouse models of sporadic AD, suggesting its protective role in the response to infections [7,8]. Furthermore, interactions between Aβ peptides and the HSV-1 fusogenic protein gB lead to the impairment of HSV-1 infectivity via preventing the virus from fusing with the plasma membrane, suggesting the antiviral activity of Aβ as an initial protective mechanism [9]. For more information on this subject, the reader is directed to a variety of specialized reviews [10,11].

There is solid evidence that neuroinflammation is a major contributor to amyloid plaque deposition. There are several ways in which neuroinflammation is thought to be linked to the formation of amyloid peptides. For example, activated microglia show increased amyloid precursor protein (APP) expression, leading to a higher level of amyloid peptides [12]. Furthermore, enzymes involved in the pathological processing of APP, such as β-secretase (BACE1) can be upregulated by pro-inflammatory signaling involving NF-kb [13]. In addition, inflammatory processes can lead to the production of reactive oxygen and nitrogen species (ROS/RNS) which can modify and crosslink Aβ, leading to the insolubility and deposition of amyloid aggregates [14]. It is important to note that the relationship between neuroinflammation and amyloid formation is complex and bidirectional. Amyloid peptides themselves can also trigger an immune response, further amplifying the inflammatory processes in the brain. In diseases like Alzheimer’s, there is a vicious cycle of neuroinflammation and amyloid pathology, which is thought to contribute to the progression of the disease [15]. Neuroinflammation has also been implicated in the progression of tau pathology and tangle formation with a multitude of different pathways being involved, and the reader is directed to specialist reviews [16,17,18,19].

While the exact triggers of neuroinflammation in AD remain under discussion, recent evidence suggests that infectious agents, such as viruses, fungi, and bacteria, may play a role in causing neuroinflammation and the accumulation of senile plaques and neurofibrillary tangles. More recently, even COVID-19 has also been linked to cognitive decline in infected patients [20]. The olfactory system presents an intriguing target for investigation due to its proximity to the brain and its early involvement in AD progression.

This paper advances the hypothesis that neuroinflammation in AD might be—at least partially—caused by pathogens gaining access to the brain through the olfactory system. It also suggests the hypothesis that rhinotillexomania (the act of picking one’s nose) might be an involuntary human movement [21] that unintentionally facilitates access of harmful pathogens to the CNS.

## 2. Pathogens in the Brain of AD Patients 

Several pathogens have been associated with AD, including viruses (e.g., herpes simplex virus type 1—HSV-1), bacteria (e.g., *Chlamydia pneumoniae*, spirochetes, treponemes) [22,23,24], fungi such as *Candida albicans* [25], and parasites such as *Toxoplasma gondii* [26]. These pathogens are known to establish persistent, latent, or chronic infections in peripheral tissues, including the nasal epithelium, where they may persist for extended periods without causing overt symptoms, until they enter the brain with pathological consequences (Figure 1). 

### 2.1. Viruses 

#### 2.1.1. Herpes Simplex Virus 1

Herpes simplex virus 1 (HSV1) is a common viral infection that primarily affects the orofacial region. It belongs to the Herpesviridae family and is a double-stranded DNA virus [28]. HSV-1 is highly prevalent globally, with a significant percentage of the population being infected. It is estimated that around two-thirds of the world’s population carry the virus [29,30,31,32]. HSV-1, responsible for causing oral herpes infections, typically manifests as cold sores or fever blisters on or around the lips; it can also cause genital herpes through oral–genital contact [33]. 

HSV-1 infections are more common in childhood and adolescence, and the virus tends to persist in the body for life after the initial infection [34,35,36,37]. HSV-1 can infect and replicate in nerve cells [38,39,40,41,42,43]. In the brain, HSV-1 remains latent (dormant) after initial infection, potentially leading to reactivation episodes [44,45]. During reactivation, the virus could cause chronic inflammation and neuronal damage, which might contribute to the development of neurodegenerative diseases, including AD [46]. 

One of the earliest studies that analyzed the presence of HSV-1 in Alzheimer’s disease brains using in situ hybridization did not find any evidence for the presence of HIV-1 in this disease; however, these negative results might have been most likely been attributed to the low sensitivity of the method [47]. 

However, further studies led by the team of Professor Ruth Itzhaki, who has been researching the potential role of HSV-1 in AD for decades, discovered that HSV-1 DNA is present in the human brain in a high percentage of older people [48,49]. A different study using next-generation RNA sequencing (RNA-seq) datasets and DNA samples extracted from AD and non-AD control brains could also detect no difference between AD brains over controls using either method [50]. 

Further studies showed that the HSV-1/AD connection is more complicated and depends on the Apolipoprotein E (APOE) genotype. APOE is a protein that plays a significant role in lipid metabolism and transport in the body, including the brain [51,52]. APOE4, in particular, is associated with an increased risk of developing late-onset AD [52,53,54]. It turned out that the HSV-1 confers a strong risk of AD in ApoE4 carriers [55]. 

Subsequently, a plenitude of biological evidence has emerged for a relationship between HSV-1 and AD, all supporting the association of HSV-1 in the brain with AD in APOE4 carriers: pathology correlations using neural stem cells (beta-amyloid production and tau phosphorylation), the preventive effects of HSV-1 with antivirals before the onset of dementia, the decreased risk of dementia following vaccination with a variety of vaccines and the association with a decreased frequency of HSV-1 reactivation shown in some vaccine studies, and anti-HSV serum antibodies [25,56,57,58,59]. ApoE4, in general, can increase or decrease the risk of viral infection. ApoE is dramatically involved in the life cycle and pathogenesis of viral infections, such as chronic hepatitis C virus (HCV), hepatitis B virus (HBV), herpes simplex virus type-1 (HSV-1), and HIV infections [60,61,62,63].

Epidemiological evidence also supports a link between HSV infection and AD. One study investigated the association between HSV infection and dementia, as well as the impact of herpes medications using data from the Taiwan National Health Insurance Research Data (NHIRD) set with 33,448 subjects. Patients infected with HSV were 2.56 times more likely to develop dementia. A significantly reduced risk of developing dementia in patients affected by HSV infection was detected after anti-viral drug treatment (RR = 0.092). These data strongly suggest that the use of anti-viral drugs in the treatment of HSV infections is associated with a reduced risk of dementia [64].

In summary, while it does not appear that the percentage of HSV-1 is different between AD and control subjects, it is rather the difference in the re-activation of already present HSV-1 with subsequent neuro-inflammation which emerges as a risk factor for AD. 

#### 2.1.2. Human Herpesvirus 6 (HHV6) and Other Herpesviruses 

Other viruses of the Herpesvirus family have also been mentioned as risk factors for AD. For example, a subsequent study identified three other herpesviruses in the brain: human herpesvirus 6 (HHV6)-types A and B (Roseola), herpes simplex virus type 2 (HSV2, genital herpes), and cytomegalovirus (CMV). There is a significant overlap between the prevalence of HHV6 and HSV-1 in AD brains, with HHV6 being present in a significantly higher percentage of AD than in age-matched normal brains (70% vs. 40%). It is impossible to rule out the presence of HHV6 as an opportunist in this scenario, but it is also possible that it will increase the harm that HSV-1 and apoE4 confer in AD [50,65,66,67,68]. 

Another study points to a virus cascade, suggesting the involvement of Varicella Zoster Virus in AD via the reactivation of quiescent HSV-1. In an in vitro experiment, cells infected with VZV did not show beta-amyloid (Aβ) and phosphorylated tau (p-tau) accumulation but did exhibit gliosis and elevated levels of pro-inflammatory cytokines, indicating that VZV’s effect on AD/dementia is indirect. Surprisingly, it was discovered that VZV infection of cells that were previously quiescently infected with HSV-1 results in HSV-1 reactivation and subsequent AD-like alterations, including the accumulation of Aβ and p-tau [69,70].

No widely available vaccine for herpes simplex virus (HSV-1) has been introduced into the market yet, so the protective effects of such a vaccine on AD could not be investigated yet. Developing a vaccine for these viruses has proven to be challenging due to their ability to establish latency and reactivate periodically [71,72,73,74]. However, vaccines for shingles (a member of the herpes family of viruses that also causes chickenpox) such as Zostavax have been around for decades, and a large study hints at a link between the shingles vaccination and reduced dementia risk. An analysis of nearly 300,000 people found an association between the shingles jab and a lower rate of dementia, according to a study of health records from around 300,000 people in Wales. The analysis found that getting the vaccine lowers the risk of dementia by 20% [75].

Preventive measures against infection with Herpes and similar viruses follow the classic rules of infection control, including avoiding close contact with individuals experiencing active outbreaks, refraining from kissing or engaging in oral–genital contact during an active infection, and not sharing personal items that may come into contact with infected saliva. Additionally, proper hand hygiene has been shown essential to minimize the risk of viral transmission [76,77].

#### 2.1.3. SARS-CoV-2

Since 2019, a new virus has been suspected to contribute to the evidence of viral involvement in AD: the severe acute respiratory syndrome coronavirus 2 (SARS-CoV-2), which causes the COVID-19 coronavirus disease. There is evidence that SARS-CoV-2 has neurotropic characteristics and can enter the central nervous system (CNS). The spike protein, which is present on the surface of the virus, binds to the angiotensin-converting enzyme 2 (ACE2) receptor, allowing the SARS-CoV-2 virus to enter cells on a molecular level [78]. The blood–brain barrier (BBB), which is made up of endothelial cells, expresses ACE2 receptors, which facilitate SARS-CoV-2 invasion into the central nervous system (CNS) [12,79]. The virus can enter an organism and travel through numerous identified pathways to the CNS [79].

A hematogenous spread by infected leukocytes that move to the brain is the first suggested mechanism [80]. Another method is for the virus to enter glial cells directly from vascular endothelial cells, cross the BBB, and then enter infected neurons via transsynaptic transfer. Retrograde axonal transport of SARS-CoV-2 from the nasopharynx to the CNS via the olfactory nerve and olfactory bulb has also been reported [79].

Moreover, SARS-CoV-2 infections have been shown to frequently cause severe neurological symptoms. The prevalence of neurological symptoms was higher in patients with severe infections, according to several studies that showed between one third and two thirds of patients with this condition experienced at least one neurological symptom [81,82,83,84,85]. 

Additionally, it was discovered that COVID-19 is associated with a higher risk for neurodegenerative and psychiatric diseases, including intracranial hemorrhage, ischemic stroke, parkinsonism, dementia, anxiety disorder, and psychotic disorder, among others [86]. It was found that, after contracting SARS-CoV-2, patients with Parkinson’s disease (PD) experienced worsening motor and non-motor symptoms [87]. 

Data on the severity, susceptibility, and hospitalization of COVID-19 showed that COVID-19 may increase the risk of AD [88,89]. Other research also indicates that AD seems to be particularly associated with SARS-CoV-2 infection among neurodegenerative diseases [90,91,92,93,94]. It is important to note that research in this area is ongoing, and the long-term outcomes of COVID-19 and its complications might not be known for decades.

On top of the individual viral pathogens mentioned above, studies using biobanks have linked viral exposures and neurodegenerative disease. In this study, dementia was highly correlated with viral encephalitis, other viral diseases, viral warts, influenza, influenza pneumonia, and viral pneumonia [95]. Very interestingly, signatures for viral infection and inflammation in the proximal olfactory system were also observed in patients with familial Alzheimer’s disease subjects carrying the presenilin 1 E280A mutation [96]. 

### 2.2. Bacteria 

#### 2.2.1. *Chlamydophila pneumoniae* (Clamydia)

*Chlamydophila pneumoniae (C. pneumonia)* is a bacterium that usually infects the respiratory tract and causes pneumonia and bronchitis. It can, however, also affect the immune system and continue to exist as a persistent infection inside cells, including white blood cells. People over 60 are most susceptible to this [97,98]. 

One early study, published in 1998, found evidence of *C. pneumonia* preferentially in the brains of people with AD [22]. In another study, the presence of *C. pneumoniae* was investigated in brain samples from patients with late-onset AD and compared to non-AD control subjects. A proportion of 20/27 AD patients, but only 3/27 of the controls, were PCR-positive for Chlamydia. Transcripts from several AD patients demonstrated that the organisms were alive and metabolically active. Immunohistochemical analyses further showed that astrocytes, microglia, and neurons all served as host cells for *C. pneumoniae* in AD [99]. However, several negative studies were also published, indicating the link between *C. pneumoniae* infection and AD is somewhat tenuous [100,101]. 

Many of the sample sizes in the above-mentioned studies were quite small, making it difficult to detect significant relationships. When an extensive meta-analysis was performed, a more convincing relationship between *C. pneumoniae* infection and AD became obvious. The association was identified through a systematic search of data combined from 25 case–control studies, and a five-fold increased incidence of AD with *C. pneumoniae* infection (OR: 5.66; 95% CI: 1.83–17.51) was detected [102].

Through using monocytes to pass the blood–brain barrier, *C. pneumonia* can cause neuroinflammation in the central nervous system. An intranasal inoculation can result in CNS infection, but it can also happen weeks or months later, according to a recent mouse study (discussed in more detail in a later section). The authors demonstrated that *C. pneumoniae* can infect the olfactory and trigeminal nerves, olfactory bulb, and brain in mice within 72 h through isolating live *C. pneumoniae* from tissues and utilizing immunohistochemistry. *C. pneumoniae* increases the risk of AD pathology via infecting the central nervous system through the trigeminal and olfactory nerves. The infection could lead to the of release pro-inflammatory cytokines and chemokines that have been linked to neurodegeneration in AD [103,104]. In a further cell culture study, *C. pneumonia* infection influences monocyte gene transcript expression and pro-inflammatory cytokine production. The chemokine CCL2, as well as the cytokines IFN-β 1, IL-1 and IL-6, were significantly upregulated in infected monocytes [105]. 

#### 2.2.2. *Porphyromonas gingivalis*

*Porphyromonas gingivalis* is a Gram-negative anaerobic bacterium that plays a significant role in periodontal disease [106]. It is one of the key pathogens involved in the initiation and progression of periodontitis and contributes to the destruction of gum tissue and supporting bone. *P. gingivalis* is reliant upon proteins or peptides as nutrients. This bacterium requires heme for survival because it lacks a mechanism for the biosynthesis of heme. *P. gingivalis* is primarily prevalent in bleeding chronic periodontal lesions, where erythrocyte-derived hemoglobin serves as an abundant supply of hemoglobin [107,108].

*P. gingivalis* can produce a large amount of proteinases to degrade proteins from the host or other microorganisms to meet its special nutritional requirements. These proteolytic enzymes, known as “Gingipains”, allow *P. gingivalis* to elude the host immune response and induce tissue damage, which significantly contributes to its pathogenicity and virulence. Gingipains cleave the formyl-methionyl-leucyl-phenylanine (FMLP) receptors, which causes neutrophils to become inactive and unable to recognize the invasive pathogens. IL-1, IL-4, IL-6, IL-8, IL-12, the IL-6 receptor, interferon, tumor necrosis factor (TNF), CD4, CD8, CD14, and CD54 have all been demonstrated to be degraded and rendered inactive by gingipains [34]. As a result, vital immune system communications are interfered with.

Further studies have also shown that *P. gingivalis* can directly be linked to the hallmarks of AD. For example, it promotes the accumulation of Aβ in the brain via interacting with the amyloid precursor protein (APP) and enhancing its processing into Aβ. This, in turn, leads to the formation of neurotoxic Aβ aggregates and contributes to the neurodegenerative process [109,110,111].

#### 2.2.3. *Staphylococcus aureus*

*Staphylococcus aureus* is a Gram-positive facultatively anaerobic bacterium that can be found in association with the human gut, skin, and nasopharynx. While many people are asymptomatic carriers, *S. aureus* can cause a wide range of pathologies, from mild skin and soft tissue infections to more serious conditions, such as bacteremia, pneumonia, and sepsis. In this regard, strains of *S. aureus* variously encode a variety of virulence factors that allow it to evade, manipulate, and disrupt the host immune system, cause tissue damage, and travel to other body sites (from the nasopharynx, for example); note that the latter is facilitated, at least in part, by its ability to survive inside phagocytic cells [112]. Furthermore, *S. aureus* cells readily develop resistance to antimicrobial therapy through the actions of mobile genetic elements [113,114], and it is for these reasons that it is considered an important human pathogen including for neurological diseases [115,116,117,118].

In the context of cognitive decline, various studies have linked *S. aureus* to AD pathology. For example, *S. aureus* has been shown to bind to amyloidogenic peptide Aβ1–42 and accelerate its agglutination [119]. Additionally, Lim et al. (2022) demonstrated the presence of *S. aureus* around or imbedded in Aβ plaques (in human tonsillar tissue), and in human tonsil and brain organoids, *S. aureus* increased Aβ levels [120]. Furthermore, it has recently been shown that a peripheral *S. aureus* infection, which was achieved via nasal inoculation, aggravated amyloid pathology and inflammation around plaques in an AD mouse model [121]. 

### 2.3. Fungi

*Candida* spp., *Malassezia* spp., *Cladosporium* spp., and *Alternaria* spp. are common fungi detected in the brain of patients with AD. These fungi are opportunistic organisms and can cause infection in immunocompromised patients [3,122]. Because of its widespread proliferation as a commensal fungus, extensive variety of fungal morphologies, and predominate proliferation in AD tissues, *Candida albicans* has been the subject of the majority of studies as the template of human fungal pathology [123]. Plenty of histological evidence indicates that the central nervous system (CNS) of AD patients contains *Candida albicans*, including fungal cells and hyphae. Several areas of the brain, including the choroid plexus, entorhinal cortex/hippocampus, external frontal cortex, and cerebellar hemisphere, contain fungi that are not present in control subjects’ brain tissue [124]. Interestingly, the Aβ peptide (major content of amyloid deposits) exhibits antimicrobial and antifungal activity and shows particularly strong inhibitory activity against *Candida albicans*, indicating that plaque formation is a side effect of a pro-inflammatory anti-fungal host response [125]. In a further, more comprehensive study, next-generation sequencing (NGS) was used to identify fungal species in the brains of AD patients using the Illumina platform. All four CNS regions under study shared *Cryptococcus curvatus* and *Botrytis cinerea*. Eight more AD patients’ entorhinal/cortex hippocampus samples underwent extensive investigation, revealing a range of fungi, some more prevalent than others. The five genera Alternaria, Botrytis, Candida, Cladosporium, and Malassezia were shared by all nine individuals. The authors of this study suggest that the differences found between the fungal species in each patient may explain the differences in the clinical symptoms in their progression of AD [126].

Fungi typically enter the brain through systemic infection as a result of epithelial barrier disruption brought on by skin and gut colonization. Potentially, fungi, particularly the Candida species, can enter the brain and cause amyloid precursor protein (APP)-accumulating fungal glial granulomas. 

In summary, there is considerable evidence for the contribution of fungi and other previously mentioned pathogens to the development of AD. The “infectious hypothesis of AD” proposes that certain pathogens, once they enter the brain, might trigger an immune response that leads to inflammation and damage to neurons, contributing to or accelerating the characteristic (neuro-inflammatory) symptoms of AD as well as being responsible for the formation of the pathological hallmarks, senile plaques and neurofibrillary tangles [25,127]. In analogy, one could describe the pathogen-mediated contribution to neuroinflammation in AD as “death by a thousand bugs”. 

## 3. The Olfactory System and Alzheimer’s Disease

The olfactory system is responsible for the sense of smell and comprises the olfactory epithelium in the nasal cavity, the olfactory bulb, and several olfactory-related brain regions. The first link of olfaction to AD risk came from epidemiological studies where impaired olfaction was linked to a more than six-fold higher risk for cognitive impairment in older adults over 5 years [128].

Interestingly, carriers of the gene variant APOE e4 lost their ability to detect odors earlier than those without it [129,130]. The olfactory bulb is one of the first brain regions affected by AD pathology, exhibiting Aβ deposition and tau hyperphosphorylation in early disease stages [131,132,133,134,135]. The early involvement of the olfactory system in AD has led researchers to explore its potential role as a possible gateway for pathogens into the brain as early as 1988 [136,137,138].

The olfactory system provides an anatomical pathway for pathogens to enter the brain. This pathway is characterized by olfactory sensory neurons projecting their axons through the cribriform plate, a portion of the ethmoid bone located at the base of the skull, directly connecting the nasal cavity to the olfactory bulb. The cribriform plate, while essential for olfaction, also represents a potential route for pathogens to invade the brain [139,140]. 

The brain is usually well protected against microbial invasion by cellular barriers, such as the blood–brain barrier (BBB) and the blood–cerebrospinal fluid barrier (BCSFB) [141]. In addition, cells within the CNS are capable of producing an immune response against invading pathogens. Moreover, the integrity of the BBB in the olfactory bulb is relatively weaker than in other brain regions, making it more susceptible to pathogen infiltration [142,143,144,145,146,147,148]. Disruption of the BBB could allow pathogens to bypass the brain’s immune defenses and trigger neuroinflammation. Bacteria, amoebae, fungi, and viruses are capable of CNS invasion, with the latter using axonal transport as a common route of infection [149,150]. Nonetheless, a range of pathogenic microbes make their way to the CNS, and some infections can cause significant morbidity and mortality [151,152,153,154,155], including those that lead to serious acute infections such as bacterial meningitis [151,154].

In the following sections, we provide more evidence that those infections leading to neurodegenerative processes including those in AD are caused by chronic or repeated exposures (death by a thousand cuts), and in the following sections, we propose that nose picking is an under-investigated route of transmission of pathogenic microorganisms onto the nasal mucosa, facilitating their transport into the brain. 

## 4. A Potential Role of the Oral Microbiota in AD? 

The gastrointestinal tract begins in the oral cavity, which harbors diverse and dynamic communities of microorganisms commonly referred to as the oral microbiota [156]. The oral microbiota is comprised of hundreds of bacterial species (nearly 800 species), Archaea, fungi, viruses, and protozoa, organized in biofilms on mucosal and tooth surfaces [157,158]. The bacterial communities in the saliva account for about 70% of the oral microbiota, with genera including Streptococcus, Prevotella, and Veillonella [159]. Recently, the oral microbiota has emerged as the key player in the maintenance of health throughout the lifetime of the individual, as well as in the occurrence, progression, and treatment of different diseases that are not just limited to the oral cavity [158]. 

Several studies in the current literature have demonstrated the link between the oral microbiota and neuropsychiatric and neurodegenerative disorders including AD, potentially through the oral microbiota–brain axis [160,161,162]. 

A periodontal infection due to the dysbiosis (disturbed microbial homeostasis) of the oral microbiota, resulting in inflammation and pathogen-induced toxin formation, has been increasingly associated with a greater risk for AD, with some studies arguing a causal relationship [161]. Several pathogens, such as *Porphyromonas gingivalis*, *Treponema denticola*, *Tannerella forsythia*, and *Aggregatibacter actinomycetemcomitans,* can cause chronic periodontitis [156]. Of these pathogens, *P. gingivalis* has been considered a primary driver with a potential causative relationship with AD pathogenesis [156,161,162]. Furthermore, other periodontal pathogens, including *Fusobacterium nucleatum*, *Prevotella intermedia*, *Actinomyces naeslundii*, and *Eubacterium nodatum,* have been highlighted as potential biomarkers/risk factors for the early diagnosis of AD as antibodies to most of these periodontal pathogens were detected in subjects years before cognitive impairment [156,163,164].

In addition to periodontal pathogens, the altered composition of the oral microbiota has also been reported in participants with AD in several studies. For instance, a higher abundance of Moraxella, Leptotrichia, Sphaerochaeta, Lactobacillales, Streptococcaceae, Firmicutes/Bacteroidetes, and a significant reduction of Rothia and Fusobacterium was observed in the oral microbiota of AD participants [165,166,167]. Interestingly, similar to *P. gingivalis*, other Gram-negative bacteria—Moraxella, Leptotrichia, and Sphaerochaeta—found in the oral cavity were linked to Aβ plaque formation through the production of LPS in the brains of AD patients [166,168]. Similar associations among LPS produced by Gram-negative E. coli K99, amyloid plaques, and AD were previously established in both humans and rats [169]. 

## 5. Conversion of a Symbiotic Nasal Microbiota to a More Pathogenic Phenotype—A Potential Role for AD Pathogenesis? 

During adulthood, the composition of the nasal microbiota remains relatively constant and mostly undergoes changes in middle age with the bacterial communities *Staphylococcus*, *Streptococcus*, *Veillonella*, *Cutibacterium*, *Corynebacterium*, *Lactobacillus*, *Moraxella*, *Fusobacterium*, *Mycoplasma*, *Pseudomonas*, *Haemophilus*, *Neisseria*, *Escherichia,* and *Prevotella* dominating the nasal microbiota of healthy adults (between 40–65 years) [20,170]. With aging, the composition of the nasal microbiota changes and becomes similar to that of the oropharyngeal microbiota [20]. These microbes influence olfactory function, and disturbances to the normal nasal microflora can potentially contribute to the pathogenesis of neurodegenerative diseases such as AD [20,170]. The bidirectional associations between the nasal microbiota and AD have been recently underlined through signifying its potential involvement in both the pathogenesis and treatment of AD [170]. However, the comprehensive profiling of nasal microbiota in AD patients has been limited, with most studies centered on *C. pneumonia*. As mentioned in the previous section, the obligate intracellular pathogen *C. pneumonia* can disrupt the nasal microbiota and initiate the development of AD [171]. The theory of the nose-to-brain axis mediated through bacterial transportable toxins in neurodegenerative diseases such as AD and multiple sclerosis has been postulated before [171,172]. 

Despite the significant advancement in next-generation sequencing techniques and microbiome research overall, the exact association between the oral and nasal microbiota and AD is still speculative. More clinical and epidemiological studies in conjunction with comprehensive microbiome profiling are prudent to distinguish between the cause vs. correlation aspects of the oral and nasal microbiome with AD pathology. Further animal studies will also provide mechanistic insights into understanding the fundamental role of the oral and nasal microbiota in the diagnosis, development, progression, and clinical outcome of AD. 

In summary, we suggest that the conversion of a symbiotic to a pathogenic nasal microbiome might be a worthwhile hypothesis to put forward as an additional risk factor for AD. 

## 6. Animal Experiments Link Infection through the Olfactory System to AD Pathology 

Animal experiments further strengthen the hypothesis that pathogens entering the body through the olfactory and trigeminal nerves might be directly linked to the pathology of AD. A variety of studies have used *Chlamydia pneumonia*, the bacteria that causes respiratory tract infections but can also infect the central nervous system (CNS) and contribute to late-onset dementia [173]. 

In one of the most crucial and groundbreaking studies, mice were infected intranasally with *C. pneumoniae*. The olfactory bulb and the trigeminal and olfactory nerves showed the presence of the pathogen within 72 h. *C. pneumoniae* was able to infect peripheral nerves and CNS glia. Furthermore, at 7 and 28 days following inoculation, the *C. pneumoniae* infection caused the deregulation of important genes related to the folding of proteins and aiding in protein aggregation, processes that are implicated in the etiology of AD [103]. Most convincingly, Aβ accumulations, a characteristic hallmark of AD, were also detected adjacent to the *C. pneumoniae* inclusions in the olfactory system [103]. This study directly links the formation of senile plaques to the olfactory inoculation of a pathogen. 

## 7. Pattern Recognition Receptors—A Molecular Link from Invading Pathogens to Neuroinflammation 

One of the mechanistic links between pathogens and cells involves pattern recognition receptors (PRRs). They play a pivotal role in the intricate web of the immune response, linking them closely to the process of inflammation. PRRs are specialized receptors expressed by immune cells, which enable the recognition of distinct molecular patterns that are often associated with microbial pathogens or damaged host cells [174]. This recognition triggers a series of events leading to inflammation, a crucial defense mechanism against infections and tissue injury. When PRRs encounter pathogen-associated molecular patterns (PAMPs) or danger-associated molecular patterns (DAMPs), they initiate signaling cascades that activate immune cells such as macrophages and dendritic cells [175]. These cells release pro-inflammatory cytokines, including interleukins and tumor necrosis factor, which orchestrate the recruitment and activation of other immune cells to the site of infection or injury [175]. This influx of immune cells leads to local vasodilation and increased blood flow, causing the redness and heat which are characteristic of inflammation [168]. Moreover, PRR activation induces the production of reactive oxygen species and nitric oxide, contributing to the destruction of pathogens but also potentially causing tissue damage [176,177].

PRR can also respond to endogenous ligands, referred to as damage-associated molecular patterns (DAMPs) which include cytosolic DNA, uric acid, and Aβ [178,179,180,181]. While predominantly studied in immune cells, especially those of myeloid lineages, these PRRs are also expressed by both glia and neurons (Figure 2).

### 7.1. NLRP3

NOD, LRR, and pyrin-domain containing 3 (NLRP3) is a member of the Nod-like receptor (NLR) family [182,183]. NLRP3 is activated by a wide range of exogenous and endogenous ligands including LPS, β-glucans, and viral nucleic acids [184]. Once activated, NLRP3 interacts with apoptosis speck-like protein containing a CARD (ASC) and procaspase-1 to form an inflammasome which activates caspase-1 through cleaving the procaspase form. Inflammasome activation can also induce NF-κB activation and subsequent pro-inflammatory cytokine production [185]. Active caspase-1 can, in turn, cleave the pro-forms of IL-1β and IL-18 and gasdermin D, a pore-forming protein that induces lytic cell death via pyroptosis [186]. The NLRP3 inflammasome can be activated by amyloid-β, neurofibrillary tangles (NFTs) containing hyper-phosphorylated tau, and amyloid plaques [187,188]. The initial recognition of increased amyloid-β by NLRP3 can cause the polarization of microglia to an M1 phenotype, which results in greater inflammatory responses to subsequent signals, driving ongoing dysregulation of the brain microenvironment [189]. 

In general, NLRP3 activation requires two steps: a priming step which induces expression of the core proteins and an activation step where the primed cell recognizes the presence of PAMP or DAMP ligands. Induction of pro-inflammatory signaling by non-NLRP3 ligands present in pathogens introduced into the olfactory nervous system could prime NLRP3 activation, lowering the threshold to later responses to Aβ [190]. NLRP inhibitors are an interesting alternative treatment being tested for limiting AD pathology [191].

### 7.2. cGAS/STING

The presence of DNA in the cytoplasm of a cell is abnormal and seen as a ‘danger’ signal by the innate immune system. The cyclic guanine monophosphate adenosine monophosphate (cGAMP) synthetase (cGAS) and Stimulator of interferon genes (STING) pathway is one of the major sensors/responders to the presence of cytosolic DNA [192,193]. Cytosolic DNA is bound by cGAS, inducing the generation of the second messenger molecule cGAMP, a potent activator of the endoplasmic reticulum-associated STING protein [194]. STING undergoes a conformational change, inducing downstream signaling and leading to type I interferon production. The cGAS/STING pathway has recently been implicated in aging-related neurodegeneration—during aging, mitochondria have reduced integrity, leading to the release of mitochondrial DNA (mtDNA) into the cytosol, triggering cGAS/STING activation and driving neurodegeneration [195]. The accumulation of DNA damage from defective cell replication or damage via reactive oxygen and nitrogen species can also result in cytosolic DNA accumulation and cGAS/STING activation [196]. Intriguingly, cGAS can also be transferred through gap junctions to neighboring cells, resulting in cGAS/STING activation in healthy adjacent cells [193]. The phenomenon may be beneficial in the context of altering surrounding cells to infection but, in the context of inflamm-aging, may result in the propagation of immune dysregulation to a wider area of the brain. Similar to NLRP3 inflammasome activation, components of the cGAS/STING pathway are usually expressed at lower levels in healthy brain tissue, but pro-inflammatory signaling increases core protein expression, amplifying the effect of the activation of this pathway [197].

### 7.3. Toll-like Receptors (TLRs)

There have been 10 functional Toll-like receptors identified in humans. In the innate immune system, TLRs also respond to a wide range of PAMPs and DAMPs, but these receptors are also expressed by both neurons and glia [198,199]. TLR signal via adaptor proteins to induce NF-κB activation and/or type I interferon production. TLR signaling circuitry is expressed in healthy brain tissue as TLR may also play a role in brain development and homeostasis, independent of the role in pathogen recognition. In the context of AD, TLR2 and TLR4 are upregulated in areas surrounding plaques, predominantly in microglial cells, and TLR5 expression is increased in the frontal cortex of humans with AD [200,201,202]. TLR4 variants have also been associated with Alzheimer’s disease [203,204]. TLRs may play a protective role in the early stage of disease through activating the phagocytic capacity of microglia and increasing the clearance of plaques. However, in later stages of the disease or when the system becomes overwhelmed, TLR signaling is pathogenic and drives the dysregulation of the microenvironment, including microgliosis, axonal trimming, and links to other potentially pathogenic pathways. For example, TLR4 may interact with NLRP3 signaling specifically via TLR4 providing a priming signal to increase NLRP3, ASC, and caspase-1 expression in glia and enhancing inflammasome activation [190]. Type I interferon induction would also result in increased cGAS/STING expression, which could enhance the effects of aging through increasing the recognition of mitochondrial DNA and subsequent pro-inflammatory signaling.

Overall, the activation of each of these pathways, and potentially others, will likely work together to create a pro-inflammatory microenvironment that promotes the neurodegeneration seen in Alzheimer’s disease. The introduction of PAMPs from bacteria, fungi, and viruses introduced into the olfactory neural system could result in early-stage inflammation, leading to feed-forward loops which result in wider-spread inflammation in the brain and may contribute to broader impacts of peripheral/systemic inflammation in driving neurodegenerative diseases.

## 8. Rhinotillexomania (Nose-Picking)—A Widespread, Unhealthy Habit? 

The medical term describing the act of picking one’s nose is “rhinotillexomania” [21]. The first systematic scientific study of nose-picking was undertaken in 1995 by Thompson and Jefferson, with the results published in their manuscript: “Rhinotillexomania: psychiatric disorder or habit?” [205]. A survey was sent out to 1000 adult residents of Dane County (Wisconsin, USA), and 91% of their respondents confessed to picking their noses. A total of 1.2% of the respondents admitted to doing it at least once each hour [205]. In another study, Andrade and Srihari compiled data from 200 pupils in four Indian schools, and nearly all of them admitted to picking their noses, on average four times per day. When asked for the reasons, the most common answers were to relieve an itch or to clear out nasal debris. Interestingly, 12% admitted that they picked their nose because it “felt good” [206].

The human nasal cavity plays a crucial role in respiration and is lined with cilia and mucus-producing cells that help trap and expel foreign particles, such as dust, allergens, and pathogens. The accumulation of these particles can lead to the formation of dried mucus, commonly referred to as “buggers,” within the nasal passages [207]. When the accumulation becomes excessive, it can hinder the smooth passage of air through the nose, potentially leading to breathing difficulties or discomfort. Nose picking, as an instinctive behavior, is a subconscious response to relieve the discomfort associated with nasal obstruction. Through removing dried mucus and other nasal debris, individuals may experience temporary relief from breathing difficulties. This unhealthy habit can lead to a perceivable improvement in airflow, thus reinforcing the behavior as a means to achieve immediate relief. It is essential to note that the temporary relief obtained from nose picking is not a substitute for proper nasal hygiene, which involves regular cleaning and maintenance of the nasal passages through gentle methods such as saline nasal rinses or blowing the nose [205].

However, nose-picking is generally not safe, in and outside of healthcare environments. For example, nose pickers surveyed in an ENT clinic were more likely to have *Staphylococcus aureus* in their nostrils than non-pickers, according to a 2006 study. The study discovered a similar phenomenon in healthy volunteers: a positive connection between self-reported nose-picking frequency and both the frequency and amount of *S. aureus* present [208].

A case report describes an even more severe nose-picking complication: a patient developed a nasal septal defect due to repetitive nose picking, resulting in recurrent infection and inflammation of the sinuses, which subsequently led to an erosion of the medial orbital wall [209]. Another interesting case study highlights the observations above. A 66-year-old woman who had a lengthy history of nose picking and blowing arrived at the emergency room with acute delirium and several physical problems. She was then repeatedly admitted for methicillin-sensitive *Staphylococcus aureus* (MSSA) cultures due to recurrent sepsis, meningitis, endocarditis, cystitis, and discitis. The nose was assumed to be the source of infection when an intranasal swab for MSSA produced identical culture and sensitivity results as earlier blood, CSF, and urine cultures. This case aims to elucidate the harmful effects of nose-picking as well as the importance of prevention, diagnosis, and management [210].

A very recent study showed nose picking as a risk factor for COVID-19. In a cohort study among 404 healthcare workers (HCWs) in two university medical centers in the Netherlands, SARS-CoV-2 incidence was about four times higher in nose-picking HCWs compared to participants who refrained from nose-picking [211].

The first study linking nose-picking to AD was presented by Henderson et al. [212], who conducted a case–control study of Alzheimer’s disease (AD) and correlated the age of onset (which separates familial from sporadic AD) to environmental exposures that might injure the brain. They found that later-onset AD was found to be positively associated with starvation/malnutrition and nose-picking [212]. However, it is essential to understand a fundamental principle of epidemiological studies: they can only prove association, not causation.

In summary, we suggest that nose-picking increases the transfer of pathogenic microorganisms from the hand into the nose changing the nasal microbiome from a symbiotic to a pathogenic type, with possible consequences of a chronic low-level brain infection via the olfactory system, subsequent neuroinflammation, and neurodegenerative diseases including AD. 

## 9. Conclusions and Implications for the Prevention of AD 

In conclusion, our review proposes the hypothesis that neuroinflammation in AD might be partially caused by pathogens entering the brain through the olfactory system.

However, one of the limitations of the “infection hypothesis of AD” is the debate of what comes first, the chicken (AD) or the egg (infection). Is it because subjects have a compromised immune system that they develop an infection leading to neuroinflammation and AD? Or does increasing inflammation due to aging and sub-clinical AD cause immune defects or unhealthy habits allowing the entry of olfactory pathogens?

Understanding the potential role of olfactory pathogen entry in AD-associated neuroinflammation opens up new avenues for prevention. Among all the entry routes, the improvement of hand hygiene might be an easy prevention step, as learned from the COVID-19 epidemic. One of the lessons learned from COVID-19 is the value of hand hygiene through frequent hand washing and the use of hand sanitizers, and we suggest these routine hygienic procedures be mandatory routine procedures for the incurable nose-picker.

## Figures and Tables

**Figure 1 biomolecules-13-01568-f001:**
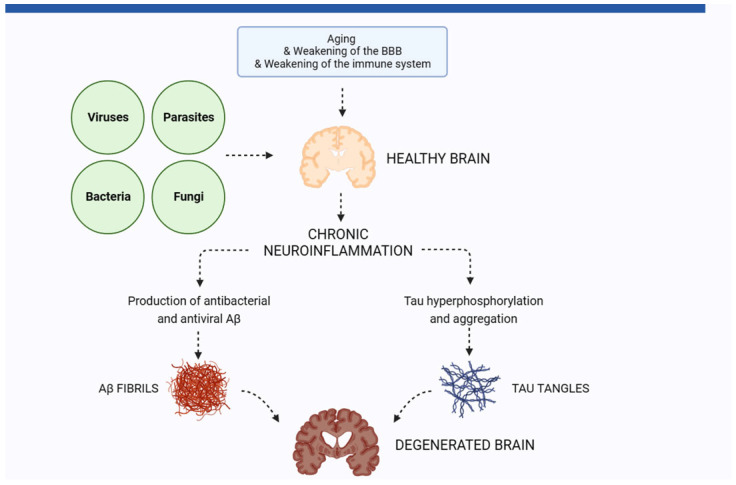
Chronic neuroinflammation as a critical player in pathogen-mediated neurodegeneration. Infectious agents entering the brain, facilitated by aging-related factors, including a weaker immune system and blood–brain barrier (BBB), lead to chronic neuroinflammation. As a reaction of the brain, β-amyloid (Aβ) is produced, which has antibacterial and antiviral properties. Aβ aggregates to fibrils, forming senile plaques, which can lead to neurodegeneration via direct and inflammatory action. Chronic neuroinflammation also leads to the formation of intracellular neurofibrillary tangles, with hyperphosphorylated tau as a precursor. Both senile plaques and tangles are hallmarks of AD, as well as pathogenic agents driving the neurodegenerative process. Image inspired by Vigasova et al. [27]. Figure created using Biorender, https://www.biorender.com, accessed on 22 September 2023.

**Figure 2 biomolecules-13-01568-f002:**
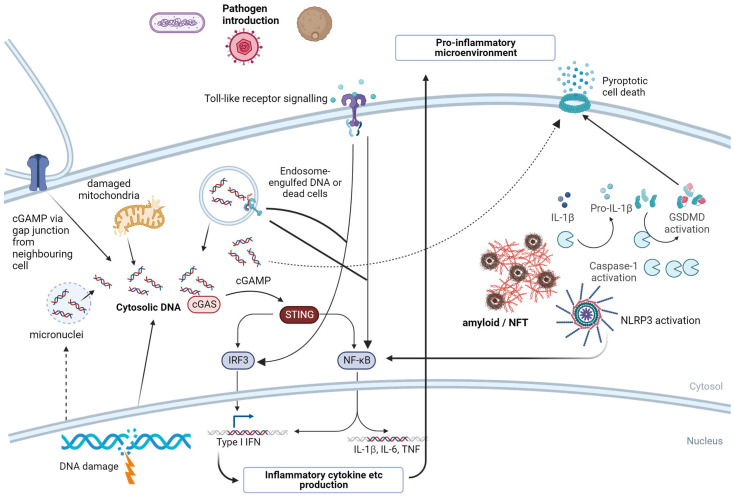
Multiple pathogen recognition pathways may be linked to Alzheimer’s disease. This figure summarizes the key pathways identified in the literature relevant to AD. Introduction of pathogens into the nasal cavity (see pathogen introduction, top left) and subsequently to the olfactory nervous system could lead to priming of PRR signaling via Toll-like receptors, NLRP3, and the cGAS/STING pathway, which induces inflammatory cytokine production and potentially pyroptotic cells death (upper right), which may in turn lead to further inflammation and greater responses to endogenous ligands such as cytosolic DNA and amyloid (bold). Figure created in Biorender. NFTs—neurofibrillary tangles; GSDMD—gasdermin; IRF—interferon regulatory factor; IL—interleukin; TNF—tumor necrosis factor. Figure created in Biorender, https://www.biorender.com, accessed on 22 September 2023.

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
