# Peer review of "Neuroinflammation in Alzheimer’s Disease: A Potential Role of Nose-Picking in Pathogen Entry via the Olfactory System?"

_biomolecules, 2023, doi:10.3390/biom13111568_

Round 1
Reviewer 1 Report
This manuscript by Zhou et al., discussed the role of nose-picking pathogens in the pathogenesis of Alzheimer’s disease (AD) by reviewing previous literatures. They picked up several representative pathogens, such as HSV1, HHV6, C. pneumonia, and P. gingivalis, and discussed the role of olfactory systems, and pathogen receptors, such as NLRP3, cGAS/STING, and TLRs. These topics were well described, and discussed in the manuscript. References also appears to be appropriate.
A few minor points:
1. As they described, COVID-19 might be also important players, while their description seems to be not enough. Please review recent literatures and discuss potential importance of COVID-19 in the pathogenesis of AD.
2. Pathogens could enter directly into neuron and interact with Abeta metabolism or tau aggregation. Also, such pathogens could induce inflammation through interacting with microglia or astrocytes. It is not clear which pathways are more important for AD pathogenesis. It might be dependent on type of pathogen and their receptors expressed in each cell types. Please discuss this point.
Reviewer 2 Report
The authors discuss an interesting and topic in Alzheimer´s disease (AD) that is the possible of an infectious origin in the pathogenesis of AD, and propose the nasal route as an entry site of pathogens to the brain causing neuroinflammation.
Nevertheless, the review is badly structured and difficult to follow. It needs major revision to shorten it (there are several iterations), to eliminate some topics (the role of the macrobiota is not clearly related), the structure of the review should focus on their proposal giving the evidence for it more clearly.
Some of the confusing sentences:
In the abstract, "Innovative materials almost mutual mechanisms in the aetiology of cancer and AD advocates novel treatment approaches."
In the introduction lines 4-5, "Cancer can begin at any age, but once it reaches a certain age range, it typically manifests as AD."
In addition, there are several stop signs where it seems there should be a comma (there is no capital letter after it), or a stop followed by a sentence that is not related. This occurs even in the abstract.
Therefore, the review needs thorough revision of the English writing to be a contribution.
As mentioned above the quality of the english writing is very poor and need major revision.
Reviewer 3 Report
The Ms by Zhou et al discusses about the etiology of Alzheimer’s disease (AD), by reviewing the role of pathogenic microorganisms, including viruses, bacteria and fungi, in this disease. This is not novel as hundreds of reviews focused on this aspect can be found in the literature. Indeed, authors state in the Figure 1 legend that the figure has been “inspired by Vigasova et al [13]”.
The Ms also suggests that the etiology of AD could be favored by nose picking, a behavior that favors pathogen entry via the olfactory system. This concept is not novel since a positive association between late-onset AD and nose picking was described by Henderson et al. (Psychol. Med. 22: 429-436, 1992). Surprisingly, this paper has not been included among the references.
Another important issue is that many statements throughout the main text (including sections 2.1.1, 3, 7, 7.1, and 7.2) lack references. This lack of references is unacceptable for a review paper.
Minor points
1. line 102: “next generation RNA sequencing” instead of “RNA sequencing”
2. Line 142: "(hallmark of AD)" should be removed since this has already been discussed in the first heading (The Role of Neuroinflammation in Alzheimer’s Disease).
3. Line 158: “classic rules” instead of “classicrules”.
4. Line 169: “The first study of this kind”. Can authors elaborate on the meaning of “this kind”.
5. Line 178: “Many of the studies were too small”. Do authors mean that the sample size was too small in these studies?
6. The information included in lines 246-248 should be transferred to the first heading (The Role of Neuroinflammation in Alzheimer’s Disease).
7. Lines 335-336: “Xie, […] Zhang” should be removed.
8. Line 364: “important pathways”. Which ones?
9. Line 367: Why do authors state that the mentioned study “received international attention”? All papers with citations do attract international attention… What makes the difference?
10. Line 371: “pattern” instead “(Pattern”.
Reviewer 4 Report
This manuscript is very well-written and provides a comprehensive review of the infectious hypothesis of Alzheimer’s disease, most specifically regarding why the olfactory tract would represent the most likely point of entry for viral and bacterial pathogens into the brain. This caused by poor hand hygiene and nose-picking.
However, some aspects would deserve a deeper discussion as they represent significant, novel findings. This is the case for ApoE4, which can increase or decrease the risk of viral infection. DOI:https://doi.org/10.1016/j.omtn.2023.07.031
Also, it would be nice to expand on the viruses that we know enter mostly by the nose, e.g. influenza, Covid, RSV, etc and their influence on dementia risk. For example, see doi.org/10.1016/j.neuron.2022.12.029
Furthermore, the authors could add seminal as well as recent work on this subject including but not limited to studies like : https://doi.org/10.1016/j.neurobiolaging.2022.12.004
Lastly, limitations to this hypothesis should be presented. Which came first, the chicken or the egg? Is it because people have a compromised immune system that they develop an infection leading to inflammation and AD or is it that increasing inflammation due to sub-clinical AD causes immune defects allowing the entry of pathogens? Is the amyloid pathology a response to infection that went wrong?
Round 2
Reviewer 2 Report
I apologize since I made a mistake in some corrections (they were part of another manuscript reviewed at the same time).
However, there are some minor suggestions that I believe will improve the manuscript.
1) In line 124 the sentence "the potential involvement of HHV6 in AD" does not correspond to the section 2.1.1. which is discussing HSV-1. It should be in the next section 2.1.2.
2) TLRs, NLRP3, and cGAS/STING are mentioned (line 463) before their definition.
3) There is a stop sign instead of a comma in line 600: “via the olfactory system. subsequent neuroinflammation, and neurodegenerative diseases”
They are mentioned in the previous paragraph
Author Response
Comments |
Changes |
1) In line 124 the sentence "the potential involvement of HHV6 in AD" does not correspond to the section 2.1.1. which is discussing HSV-1. It should be in the next section 2.1.2.
|
The sentence has been removed. |
2) TLRs, NLRP3, and cGAS/STING are mentioned (line 463) before their definition.
|
TLRs, NLRP3, and cGAS/STING is removed, as the details will be discussed later. And they will be defined |
3) There is a stop sign instead of a comma in line 600: “via the olfactory system. subsequent neuroinflammation, and neurodegenerative diseases”
|
Corrected. |
Reviewer 3 Report
The authors have satisfactorily addressed all my concerns. The Ms. is now ready for publication.
Author Response
The authors have satisfactorily addressed all my concerns. The Ms. is now ready for publication.
Thank you.